# Clinical Pharmacy Activities Documented (ClinPhADoc): Development, Reliability and Acceptability of a Documentation Tool for Community Pharmacists

**DOI:** 10.3390/pharmacy7040162

**Published:** 2019-12-02

**Authors:** Nour Hamada, Patricia Quintana Bárcena, Karen Alexandra Maes, Olivier Bugnon, Jérôme Berger

**Affiliations:** 1Institute of Global Health of Geneva, University of Geneva, 1202 Geneva, Switzerland; 2Community Pharmacy of the Center for Primary Care and Public Health (Unisanté), 1011 Lausanne, Switzerland; patricia.quintanabarcena@unisante.ch (P.Q.B.);; 3Pharmaceutical Care Research Group, University of Basel, 4056 Basel, Switzerland; 4Institute of Pharmaceutical Sciences of Western Switzerland, 1011 Lausanne, Switzerland

**Keywords:** community pharmacy, documentation system, drug-related problems, pharmaceutical intervention, validation

## Abstract

Documentation of community pharmacists’ clinical activities, such as the identification and management of drug-related problems (DRPs), is recommended. However, documentation is not systematic in Swiss community pharmacies, and relevant information about DRPs, such as consequences or involved partners, is frequently missing. This study aims to evaluate the interrater and test-retest reliability, appropriateness and acceptability of the Clinical Pharmacy Activities Documented (ClinPhADoc) tool. Ten community pharmacists participated in the study. Interrater reliability coefficients were computed using 24 standardized cases. One month later, test-retest reliability was assessed using 10 standardized cases. To assess the appropriateness, pharmacists were asked to document clinical activities in their own practice using ClinPhADoc. Acceptability was assessed by an online satisfaction survey. Kappa coefficients showing a moderate level of agreement (>0.40) were observed for interrater and test-retest reliability. Pharmacists were able to document 131 clinical activities. The good level of acceptability and brief documentation time (fewer than seven minutes) indicate that ClinPhADoc is well-suited to the community pharmacy setting. To optimize the tool, pharmacists proposed developing an electronic version. These results support the reliability and acceptance of the ClinPhADoc tool.

## 1. Introduction

Clinical activities in pharmacies involve pharmacists providing patient care to optimize medication therapy and to promote health, wellness, and disease prevention [1]. Among such activities are the identification, prevention, and resolution of drug-related problems (DRPs) [2,3]. A DRP is defined as an event or circumstance involving drug therapy that actually or potentially interferes with desired health outcomes [4]. DRPs can induce negative health and financial consequences. Stark et al. have estimated that DRPs may have accounted for 816 million euros of health care expenditures over one year in Germany for 2.14 million ambulatory patients; with 80% of expenditures related to hospitalizations [5]. To prevent such consequences, DRP management needs to be optimized.

The management of DRPs involves different clinical activities, mainly determined by the setting (hospital or community pharmacy). Likewise, the documentation of such activities is performed differently depending on the setting [6]. It has been recommended that pharmacists document any professional activities that are intended to ensure the safe and effective use of drugs and that may affect patient outcomes [7]. However, documentation of clinical activities still remains a challenge [6], particularly in community pharmacies [8].

Community pharmacists face different barriers to documenting their clinical activities, but the lack of standardized documentation systems adapted to the workflow inside pharmacies in particular presents an obstacle [9]. The existing documentation tools have been deemed incompatible with the workflow in community pharmacies. Among the reasons for such incompatibility, the following have been cited: the tools’ complexity; the omission of the actions taken by the pharmacist to resolve the DRP; a greater focus on the classification of the DRP rather than on the pharmaceutical intervention and its clinical significance; or inclusively, the time consumed to complete documentation [9,10,11,12]. Furthermore, some clinical activities in community pharmacies often remain unacknowledged since the main role of community pharmacists has been drug dispensing [13]. However, the increasing involvement of community pharmacists in patient care makes it necessary to identify the clinical activities they perform for the management of DRPs [14,15]. In Switzerland, the legal framework of pharmacists’ activities has evolved since 2001 [16]. Currently, the clinical activities in the community pharmacy legally acknowledged include basic cognitive services such as delivery, counseling services, prescription/dosage/drug-drug interaction checks, or completing patients’ records. Remuneration for such clinical activities is not linked to the price of the drug, but to a fee-for-service [17]. However, the documentation of such activities is not performed on a regular and structured basis in Swiss community pharmacies. Likewise, information about the DRP management process, DRP consequences, or involved partners (other than pharmacists and patients) is frequently missing.

Since 2008 in the Community Pharmacy of the Center for Primary Care and Public Health (Unisanté), University of Lausanne (Switzerland), a tool called Clinical Pharmacy Activities Documented (ClinPhADoc), based on a previously published coding system for the documentation of clinical activities related to DRP management [18], has been used by pharmacists to document the two main phases related to these clinical activities: the detection and the management of DRP. For example, a sample based documentation for predefined periods, at the Unisanté pharmacy covering 1248 prescriptions in 2017 and 1014 prescriptions in 2018 showed 303 and 231 clinical activities related to the management of DRP, respectively [19]. However, this practice tool has not been updated or validated.

The aims of this study were: (1) to update the ClinPhADoc tool for the documentation of clinical activities in the community pharmacy, and (2) to evaluate its interrater and test-retest reliability, appropriateness and acceptability by community pharmacists. This study was conducted in collaboration with the Pharmaceutical Care Research Group of the University of Basel, Switzerland [20].

## 2. Materials and Methods

The Swiss health research authority in the Canton of Vaud (Commission cantonale d’éthique de la recherche sur l’être humain-CER-VD) determined that, because the goals of this project are focused on the professional opinion of pharmacists regarding the use of the tool and as only fictional patients’ cases are used, it is deemed to be outside the application of the Swiss law on human research.

### 2.1. Update of the ClinPhADoc Tool

The ClinPhADoc tool was based on a coding system to report and assess community pharmacists’ interventions [18] used at Unisanté since 2008. The ClinPhADoc tool defined three categories of DRPs (derived from the aforementioned coding system [18] and a previous study on DRP causes [21]): (1) clinical (managed uniquely by the pharmacists, it can affect the efficacy or the toxicity of a medication); (2) technical (managed by the pharmacists or in collaboration with the pharmacy technicians, it is related to medication use); and (3) procedural (managed by pharmacists only or in collaboration with the pharmacy technicians, it is related to renewals of outdated prescriptions authorized by the Swiss pharmacy practice law).

The ClinPhADoc also features four different steps: (1) consequence of clinical DRP (increased safety issues or lack of efficacy); (2) status of DRP (manifest or potential); (3) clinical decision of the pharmacist (modified prescription or not); and (4) partners involved in the clinical decision.

In 2015, this content was updated to reflect current practices. For this update, three databases (PubMed, Web of Science, and Embase) were consulted to identify tools: (1) used in a community pharmacy, (2) published between 2004 and 2015 (to comprise the data published during the ten years prior to this study), and (3) available in French, English, German, Arabic, or Spanish. The original documentation tool was modified using the consulted literature to constitute the final version of the ClinPhADoc tool.

### 2.2. Validation of the ClinPhADoc Tool

A prospective observational study was conducted in the regions of Geneva and Lausanne, Switzerland, from April 1 to May 15, 2015. Volunteer participants were recruited based on an announcement among the local society of community pharmacists. Twenty community pharmacists expressed their interest, and ten were randomly selected for inclusion in this study. For this study, participants had to complete online training before the start of the study. The training consisted of a document explaining the two different phases (detection and management of DRP) for documenting the clinical activities of community pharmacists using the ClinPhADoc, the four different steps, the three categories of DRPs as defined by the ClinPhADoc tool (clinical, technical, and procedural) and the 39 types of DRPs. Moreover, documentation rules enhancing the homogeneity of documentation were indicated, such as pharmacists having to document their interventions using a case reporting form, using one case reporting form for each intervention, filling in every step of the tool, and making one choice for each step except for step 4, where multiple partners could be involved in the clinical decision. A user descriptive manual was also made available to participants.

#### 2.2.1. Interrater Reliability

The 10 pharmacists used the ClinPhADoc to document 24 standardized clinical cases adapted to the community pharmacy context from Ganso et al. [22]. The documented cases that did not fit the documentation rules and the user descriptive manual were excluded. Fleiss’s kappa index (κ) was calculated for each step (1 to 4) of the ClinPhADoc using Stata (StataCorp, Houston, TX, USA) version 13. Fleiss’s kappa index was also calculated for the DRP category (clinical, technical, and procedural) to assess the agreement between raters. The kappa index is one of the most commonly used statistics to test interrater reliability [23]. Landis and Koch estimate the kappa value as fair if it is greater than 0.40, which implies a reliable system in clinical practice [24].

#### 2.2.2. Test-Retest Reliability

Test-retest reliability was assessed by the 10 pharmacists using 10 standardized cases one month after the interrater reliability tests. These cases were randomly selected among the 24 cases and modified to present a situation involving a similar DRP (e.g., wording, gender, and medications were changed). The cases that did not fit the documentation rules and the user descriptive manual were excluded. Fleiss’s kappa indexes for each step (1 to 4) of the ClinPhADoc and for DRP category were calculated; each participant was treated as two different raters for each step when initially documenting the 10 first versions of the cases and then the 10 modified versions.

#### 2.2.3. Appropriateness

To assess the appropriateness of the ClinPhADoc, each pharmacist was asked to document up to 15 clinical activities during dispensing procedures in their own pharmacies for two weeks; from April 13 to April 27, 2015, using the ClinPhADoc tool. In addition, a short description of the situation, the time needed to manage the DRP and to document the clinical activity and an anonymized copy of the prescription were also requested. The cases that did not respect the documentation rules and the user descriptive manual were excluded. Descriptive statistics summarized the collected clinical activity of the community pharmacists: including the proportion of involved medicines from the different groups of the Anatomical Therapeutic Chemical (ATC) Classification System; the proportion of the different categories of documented DRPs (technical, clinical, and procedural) and the proportion of the different involved partners.

#### 2.2.4. Acceptability and Feasibility

Acceptability was assessed through a satisfaction survey (11 questions) based on similar surveys [20,25,26]. The extent of agreement was assessed by a five-point Likert scale (strongly disagree, disagree, neutral, agree, and strongly agree). Questions evaluated mainly the acceptability of the tool by users, the feasibility of the use of the tool within the daily workflow and the importance of documenting the pharmacist’s clinical activities. Finally, pharmacists could give their comments about potential improvements to the ClinPhADoc.

## 3. Results

### 3.1. Update

The literature review identified 22 validated tools for the documentation of pharmacists’ clinical activity related to DRP management [9,10,25,27,28,29,30,31,32,33,34,35,36,37,38,39,40,41,42,43,44,45]. Among these, 9 were excluded for not meeting the inclusion criteria, being conceived for use in the hospital context (*n* = 3) [32,39,40], in another language (*n* = 2) [37,38], full text not available (*n* = 2) [41,43], no DRP classification specified (*n* = 1) [42], or already considered in the development of the first version of ClinPhADoc [34]. After comparison with the remaining tools, the ClinPhADoc was organized into two phases: A (DRP detection) and B (DRP management) and four steps. The steps included the following: (1a) three categories of DRP (clinical, technical, and procedural); (1b) two possible consequences of clinical DRPs (possible risk related to efficacy or safety); (2) DRP status (present or potential); (3) pharmacist’s intervention (if he/she modified the prescription or not); and (4) partners who were involved in the clinical decision making alongside the pharmacist (patient and/or caregiver and/or physician). The steps and their individual modifications are further described in Table 1. A visual representation of the steps is displayed in Figure 1. The final version of the ClinPhADoc tool is shown in Figure 2.

### 3.2. Validation

#### 3.2.1. Interrater Reliability 

Among the 240 documented cases, one was excluded for noncompliance with the documentation rules. The Fleiss’s kappa index values obtained for each step were above 0.40; for step 1a: 0.54 (95% confidence interval (CI): 0.53; 0.54); step 1b: 0.54 (95%CI: 0.51; 0.55); step 2: 0.57 (95%CI: 0.53; 0.61); step 3: 0.48 (95%CI: 0.43; 0.54); and step 4: 0.49 (95%CI: 0.47; 0.52).

#### 3.2.2. Test-Retest Reliability

The kappa values obtained for the participants ranged from 0.52 to 0.78, as shown in Table 2.

#### 3.2.3. Appropriateness

To evaluate the appropriateness of the ClinPhADoc, the participants documented 136 clinical activities related to DRP management out of the 150 activities initially requested (15 cases for each participant) due to time constraints imposed by the researchers. Among these activities, five were excluded for noncompliance with the documentation rules. The 131 documented clinical activities involved 120 prescriptions for 120 patients and 150 medications. The mean time required to manage the DRP including the documentation of the activities was 6 min and 36 s. Pharmacists were able to document DRPs in the three different categories defined by the ClinPhADoc (47% technical, 46% clinical, and 7% procedural). Documented DRPs included medications from the 14 main groups of the ATC Classification System. Clinical decision makers in the management of documented DRPs included, in addition to the pharmacists, physician (36%), patients (34%), and physician plus patients (2%). The pharmacists made clinical decisions alone in 28% of the documented cases.

#### 3.2.4. Acceptability

Each participant completed an online satisfaction survey, and the results are shown in Figure 3. When asked about potential modifications to the ClinPhADoc, the major improvement suggested by participating pharmacists was the development of an electronic version including protected storage of the documented clinical activities.

## 4. Discussion

The ClinPhADoc is a unique tool allowing pharmacists to document essential information about the clinical activities performed in their daily practice to manage DRPs. Such information includes, in addition to the type of problem, an estimation of DRP consequences (risk related to efficacy or safety) and who participated in the clinical decision. The updated version of the ClinPhADoc demonstrated good interrater reliability and test-retest reliability. Moreover, the ClinPhADoc was well accepted by participants and proved to be adaptable for documentation of different cases in community pharmacists’ daily practice.

The overall kappa index obtained for interrater reliability for the ClinPhADoc (kappa = 0.53) is similar to those obtained for a Swiss (kappa = 0.61) and a German (kappa = 0.58) documentation tools, which were judged to be reliable [20,39]. In addition, the ClinPhADoc demonstrated a good level of test-retest reliability. Among the 22 identified documentation tools, this characteristic remains undetermined, as only two studies assessed test-retest reliability. These results cannot be directly compared to those of the ClinPhADoc, as they were expressed in percentages rather than by kappa values [9,33]. As each kappa value including the level of uncertainty (as defined by their respective confidence intervals) was above 0.40 [24], the ClinPhADoc can be deemed as a reliable tool for use in community pharmacy practice.

The appropriateness of the ClinPhADoc for use in real practice is supported by the fact that a variety of clinical activities could be documented by participants. Only five (3.7%) of the 136 clinical activities were excluded for noncompliance with the documentation rules. Furthermore, the mean duration of a clinical activity, including documentation, was estimated to be 6 min and 36 s. Lack of time has been deemed a barrier to documentation in pharmacy practice. In a study also conducted in community pharmacies in the regions of Geneva and Lausanne, the mean duration of clinical activity was similar (estimated at 6 min and 48 s) [19]. In this latter study, the clinical activity and the invested time were documented by an external observer. In addition, according to the satisfaction survey, the documentation time needed for ClinPhADoc does not represent an obstacle for 8 out of 10 participants. These results show that the time needed for documenting clinical activities using the ClinPhADoc tool has been adapted to the community pharmacy setting. These results also indicate that the time measured by the participating pharmacists themselves is probably as reliable as the time measured by an external observer.

The majority of pharmacists agreed on the importance of documenting their clinical activities and on the usefulness of the ClinPhADoc. However, only 6 of 10 participants attested that they would use it to document clinical activities in their daily practices. Participants were not asked the motives behind such responses. In Switzerland, the documentation of identified DRPs in community pharmacies is not remunerated; nevertheless, clinical activities are paid (based on a fee-for-service) without being documented. A lack of remuneration is known to be a common barrier in the usual practice of community pharmacists [46]. Likewise, it has been reported that pharmacists perceive that interventions increase their workloads without being to their advantage (other than providing a sense of professional accomplishment) [47]. A decrease in the documentation of DRPs and associated interventions as a result of a lack of incentives was observed in a study conducted in 2008. This study showed that the mean rate of documented interventions decreased over the 4 weeks of the study, from 1.04% during the first week to 0.45% during the last week, in 15 of 20 pharmacies [18]. Therefore, even with an appropriate tool and with documentation that does not appear to be time consuming, documentation needs to be accomplished during a timeframe defined beforehand to maintain a steady reporting rate of pharmacists’ clinical activity.

The main improvement suggested by pharmacists, an electronic version of the ClinPhADoc, could facilitate its full implementation in community pharmacies. Because the ClinPhADoc tool has been only used in a paper version, an electronic version is currently under development and will be evaluated in future research. The use of technological devices (i.e., handheld computers) has enabled healthcare professionals to be more efficient in their work practices [48]. The use of such devices in pharmaceutical practice has improved the number and frequency of documented activities [49]. Therefore, it can be expected that community pharmacists will be able to improve the documentation of clinical activities with this electronic version of the ClinPhADoc. Nonetheless, the efficiency, appropriateness, and acceptability of an electronic version of the ClinPhADoc for different community pharmacies remains to be determined.

Documentation of clinical activities enables the establishment of monetary remuneration for DRP management. Clinical services are generally considered reimbursable only when they are necessary for the medical management of a patient and when the service provided and the patient’s response are carefully documented [7]. Currently, in Switzerland, the fee for clinical activities of community pharmacists (CHF 3.25 = USD 3.25 per prescription form and CHF 4.30 = USD 4.30 per medication) only covers the pharmacists’ basic cognitive services (e.g., delivery, counseling services, prescription/dosage/drug-drug interactions checks, and patient record keeping) [17]. A recent study [50] showed the limitations of the Swiss reimbursement scheme regarding payment for clinical activities of community pharmacists when delivering new hepatitis C antiviral medications. In addition, other clinical activities necessary for DRP management in the community pharmacy (e.g., monitoring of laboratory tests) are not remunerated. As part of the quality control initiatives established by the Swiss Pharmacists Association (pharmaSuisse) and health insurance, mystery shopping tests and administrative checks must be conducted for pharmacy clinical activities to be paid. Therefore, documentation of clinical activities may be helpful to demonstrate the value of the pharmacist’s services [6]. However, to be accountable, documentation of pharmacists’ clinical activities should meet established criteria for legibility, clarity, and completeness [9]. Thus, the ClinPhADoc may facilitate systematic documentation, which could enable the adjudication of monetary values to more clinical activities in community pharmacies.

Some limitations need to be acknowledged. First, the paper version of the ClinPhADoc tool allows the documentation of only one DRP at a time. Consequently, if several DRPs are identified, they must be documented on several pages. Second, the tool only allows the documentation of the immediate outcomes and required interventions using the information available at the moment when the pharmacist encounters the patient. If the patient is referred to a physician, it is usually not possible for the pharmacist to determine the outcome of the intervention (i.e., if the prescription changed or not). When the outcomes are unknown, pharmacists may indicate this using the “I don’t know” case in Phase B: DRP management section of the tool.

Regarding the limitations of the study, the sample size can be considered relatively small (10 pharmacists). Because reliability coefficients are dependent on the number of possible ratings, the level of agreement among raters and the difference among coefficients for hypothesis testing [51]; it is likely that a smaller sample size would lead to higher variability for the coefficients. Therefore, significant variability in kappa coefficients could be expected with this sample size. However, as shown by the observed values for the kappa coefficients and their respective confidence intervals above 0.40, a level of moderate agreement among raters was confirmed. Hence, ClinPhADoc can be considered reliable for use in clinical practice [24,52]. It is probable that with a larger sample, the variability observed in the reliability coefficients would decrease [52]. Furthermore, pharmacists were recruited on a voluntary basis rather than by random sampling; consequently, selection bias cannot be completely excluded. Their motivation and willingness to document their clinical activities are expected to be better than they would be for other pharmacists. In addition, participating pharmacists documented only clinical interventions related to the DRPs they selected, rather than systematically. Therefore, the appropriateness and acceptability of the tool observed in this study might differ when used on a larger scale in real practice.

## 5. Conclusions

The ClinPhADoc tool has been shown to be reliable and acceptable for documenting pharmacists’ clinical activities in a community pharmacy context. As suggested by participating pharmacists, an electronic version could improve its acceptability and facilitate its implementation for documenting clinical activities in daily practice in community pharmacies.

## Figures and Tables

**Figure 1 pharmacy-07-00162-f001:**
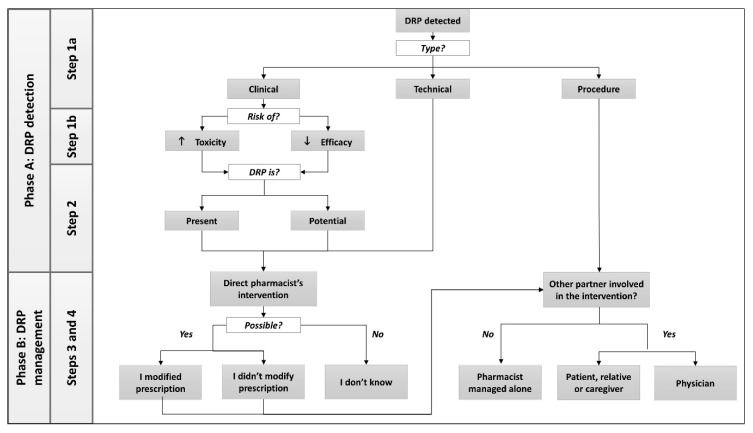
Visual representation of the steps in the ClinPhADoc tool.

**Figure 2 pharmacy-07-00162-f002:**
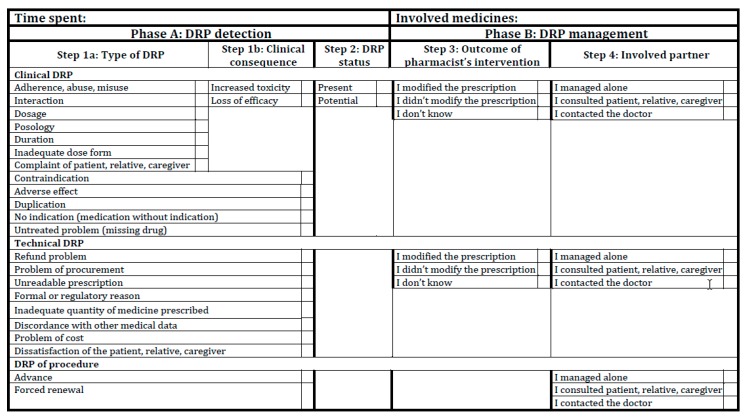
The final version of the ClinPhaDoc tool.

**Figure 3 pharmacy-07-00162-f003:**
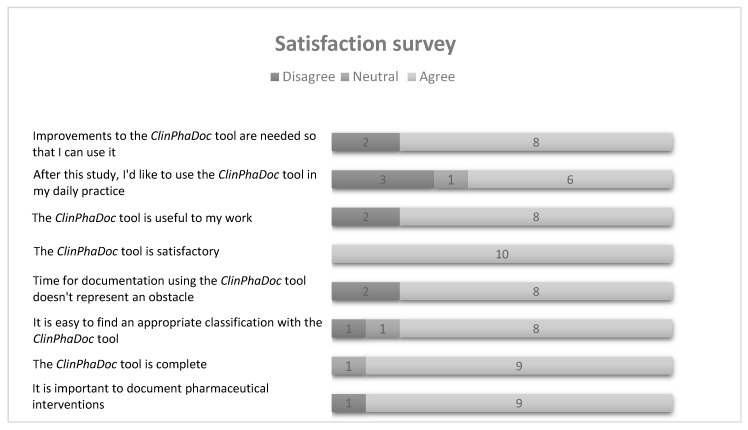
Results for the online satisfaction survey.

**Table 1 pharmacy-07-00162-t001:** Modification of the initial tool and their objectives.

Step	Modification	Objective
Step 1 DRP category, type and consequence	Split into Step 1a and Step 1b	To define the category of problem and the possible consequence (risk of inefficacy or safety issue) of clinical DRPs
	Step 1a DRP type	
	Modification of the order of DRP types	Originally considered as a DRP, the risk of inefficiency or safety issue is removed from the list of DRP types and transformed into Step 1b
	Addition of type “Complaint of patient, relative, caregiver” Addition of type “Dissatisfaction of patient, relative, caregiver”	To allow documentation of a pharmacist’s intervention related to a complaint and to avoid incorrectly documenting a DRP as “side effect”
	The type “Temporary off-trade” replaced by “Problem of procurement” The type “Incomplete prescription” replaced by “Formal or regulatory reason”	To extend the possibilities among technical DRPs
	Step 1b DRP consequence	
	The type “Indication” deleted The type “Efficacy” reworded to “Loss of efficacy” The type “Toxicity” reworded to “Increased toxicity”	To distinguish the consequences of DRPs between leading to increased toxicity or loss of efficacy
Step 2 DRP status	Step 2 “Clinical result” reworded to step 2 “DRP status” Subcategory “Present” added Subcategory “Potential” added	To determine the status of the DRP at the moment of patient encounter in the community pharmacy: present or potential
Step 3 Outcome of pharmacist’s intervention	“Modified prescription” reworded to “I [pharmacist] modified the prescription” “Unmodified prescription” reworded to “I [pharmacist] didn’t modify the prescription” The option “I [pharmacist] don’t know” added	To clarify who is involved in the intervention. Indeed, “modified prescription” could be selected if pharmacists changed the prescription alone, or in collaboration with the physician. The option “I [pharmacist] don’t know” allows pharmacists to document a clinical activity even if the outcome is unknown.
Step 4 Involved partners	Wording changed: “Relative”, and “caregiver” were also considered alongside “patient”	If the patient is not directly responsible for treatment (i.e., children or elderly patients), a caregiver or relative can also be considered as involved in the decision making

**Table 2 pharmacy-07-00162-t002:** Results for test-retest reliability. Fleiss’s kappa index obtained for each participant.

Pharmacist Identification Number (ID)	ID 1	ID 2	ID 3	ID 4	ID 5	ID 6	ID 7	ID 8	ID 9	ID 10
Kappa index (95%CI)	0.78 (0.65; 0.90)	0.58 (0.50; 0.66)	0.67 (0.60; 0.74)	0.68 (0.58; 0.74)	0.61 (0.54; 0.69)	0.53 (0.45; 0.61)	0.52 (0.44; 0.60)	0.59 (0.52; 0.66)	0.62 (0.54; 0.69)	0.57 (0.49; 0.65)

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
