# Peer review of "Clinical Pharmacy Activities Documented (ClinPhADoc): Development, Reliability and Acceptability of a Documentation Tool for Community Pharmacists"

_pharmacy, 2019, doi:10.3390/pharmacy7040162_

Round 1

Reviewer 1 Report

I find very important the topic addressed in this manuscript. Justification of clinical activities done by pharmacists is paramount in order to obtain recognition of their work as well as remuneration.

Documenting activities is complex and there are many factors that make it difficult, as to have consensus in the classifications of activities, and easy and rapid tools for doing it. This is particularly complex in community pharmacy were there is an important difficulty in the time needed for filling up records, as there is the pressure of patients/clients.

The recording of pharmacists interventions is more developed in hospital pharmacy, where it has been started before and even though the work pressure, there is more margin as you do not have the patient to attend. However, nowadays it is indispensable for the future development and progress of community pharmacy that pharmacist implement records of their activities if they want to progress in their practice.

Some things I would like to point out:

Introduction: In the paragraph (line 62-69) the authors explain the experience in Community Pharmacy in 2008 in Switzerland, and indicate that the number of prescriptions in 2017 and 2018. But, how may community pharmacies participated in this these periods? Results: Line 146. The authors indicate that 9 validated tools for documentation were excluded. What were the reasons for exclusion? Figure 1. I think that the term Adverse Effect is more suitable that Side Effect. In the ClinPhaDoc tool, I do not see the option of referral of the patient to his/her doctor. I think is one of the most habitual decisions when you cannot modify the prescription. From what I see, does the pharmacist has to fill one sheet for every DRP? How do they manage when in a treatment there is more than one PDR? Do they fill a sheet for every one?

In order to reduce the time and increase the documentation, as the authors say is basic to have an electronic version. Moreover, electronic version allows creating a database and having an easy way to exploit data and have statistics.

Discussion: Line 244. When explaining the fee for clinical activities, how these activities are checked by the payer in order to approve them and therefore be payed to the pharmacy? It would be useful that the authors explain a little in the manuscript this issue.

Author Response

October 28, 2019

Ruth Bai
Assistant Editor

Pharmacy MDPI

Subject:  Response to reviewers - Paper pharmacy_610695

We thank the editors and reviewers for their helpful comments and suggestions. In the following, the reviewers’ comments are reproduced integrally, numbered and appeared in bold text. Our response follows each comment. Any text cited from the manuscript has been put between quotation marks, while any addition to the original text is in italics; changes on response to the reviewer comments are in a redlined copy of the manuscript. Line numbers refer to those in the redline manuscript.  

Reviewer 1

(Please find attached the letter of response to the Editor and all Reviewers)

Introduction: In the paragraph (line 62-69) the authors explain the experience in Community Pharmacy in 2008 in Switzerland, and indicate that the number of prescriptions in 2017 and 2018. But, how may community pharmacies participated in this these periods?

By the reading of this comment, we believe that by “may”, the reviewer meant “many”. We meant the Community Pharmacy of Unisanté, that is one single pharmacy. We were not referring to the community pharmacies in Switzerland as a whole. To avoid misunderstandings we have reformulated the sentence as follows:

[Lines 70-78]: Since 2008 in the Community Pharmacy of the Center for Primary Care and Public Health (Unisanté), University of Lausanne (Switzerland), a tool called Clinical Pharmacy Activities Documented (ClinPhADoc), based on a previously published coding system for the documentation of clinical activities related to DRP management [17], has been used by pharmacists to document the two main phases related to these clinical activities: detection and management of DRP. For example, a sample-based documentation on predefined periods at the Unisanté pharmacy, of 1248 prescriptions in 2017 and 1014 prescriptions in 2018 showed 303 and 231 clinical activities related to the management of DRP, respectively [18].”

Results: Line 146. The authors indicate that 9 validated tools for documentation were excluded. What were the reasons for exclusion?

These tools were excluded because they did not fulfil the inclusion criteria, mainly: conceived for being used in the hospital context (n=3), other language (n=2), full text not available (n=2), no DRP classification specified (n=1) or being already considered in the first version of ClinPhADoc. This is now explained in the text as follows:

[Lines 167-171]: “Among these, 9 were excluded for not meeting the inclusion criteria, mainly: conceived for being used in the hospital context (n=3)[33,41,42], other language (n=2)[39,40], full text not available (n=2)[43,45], no DRP classification specified (n=1)[44] or being already considered in the development of the first version of ClinPhADoc[35]”.

Figure 1. I think that the term Adverse Effect is more suitable that Side Effect.

As requested the term “Adverse effect” has replaced “Side effect”. This is now visible in a reformatted Figure 1. Please see the embedded figure in the attached letter. 

In the ClinPhaDoc tool, I do not see the option of referral of the patient to his/her doctor. I think is one of the most habitual decisions when you cannot modify the prescription.

The ClinPhADoc tool was conceived to document DRPs outcomes and required interventions with the information available at the moment when the pharmacist encounters the patient. If the patient is referred to the physician, it is usually not possible for the pharmacist to determine the impact of his intervention (i.e. is the prescription changed or not). This is possible to document such situation using the “I don’t know” case in the Phase B: DRP management section of the tool. We added a comment on this point in the Discussion section:

[Lines 296-301]: “Second, the tool only allows the documentation of the immediate DRPs outcomes and required interventions using the information available at the moment when the pharmacist encounters the patient. If the patient is referred to the physician, it is usually not possible for the pharmacist to determine the outcomes of the intervention (i.e. if the prescription changed or not). When the outcomes are unknown, pharmacists may indicate this using the “I don’t know” case in the Phase B: DRP management section of the tool.”

From what I see, does the pharmacist has to fill one sheet for every DRP? How do they manage when in a treatment there is more than one PDR? Do they fill a sheet for every one?

Indeed, the pharmacists has to use one sheet by DRP, even if more than one medication is implicated. This limitation will be taken in count for the development of the electronic version of ClinPhADoc. We acknowledge this limitation in the Discussion section:

[Lines 294-296]: “Some limitations need to be acknowledged. First, the paper version of the ClinPhADoc tool allows the documentation of only one DRP at a time. Consequently, if several DRPs are identified, they should be documented in several pages”.

In order to reduce the time and increase the documentation, as the authors say is basic to have an electronic version. Moreover, electronic version allows creating a database and having an easy way to exploit data and have statistics.

We fully agree with this comment. An electronic version of ClinPhADoc is currently under development. This will be implemented and evaluated in several community pharmacies. We considered as essential to first assess the interrater and test-retest reliability on a paper version.

Discussion: Line 244. When explaining the fee for clinical activities, how these activities are checked by the payer in order to approve them and therefore be payed to the pharmacy? It would be useful that the authors explain a little in the manuscript this issue.

Fees for clinical activities of community pharmacists in Switzerland are paid per medication and per prescription. Health insurances performe an administrative check to approve them. In addition, quality control (e.g. mystery shopping tests) is regularly performed in community pharmacies by the Swiss Pharmacists Association (pharmaSuisse), in collaboration with health insurances. This is explained in the Discussion section:

[Lines 286-290]: “As part of the quality control initiatives established by the Swiss Pharmacists Association (pharmaSuisse) and health insurances, mystery shopping tests and administrative checks must be conducted for pharmacy clinical activities to be paid. Therefore, documentation of clinical activities may be helpful to demonstrate the value of the pharmacist’s services.”

Reviewer 2 Report

General comments:
There are several grammatical errors that make this manuscript very difficult to read. I pointed out only some of these errors below.
Some of the examples given are very vague.
It is difficult to validate the use of this tool with such a small sample size (n = 10).
It is unclear whether this study received ethics approval.

Specific comments:
line 38: "They" - do you mean DRPs?

line 38-40: the examples here look very vague. I see that you have referenced. Can you go in details by what you mean by negative health and financial consequences? e.g. provide some figures
Also, please elaborate what you mean by "context"

line 42-43: "to ensure the safe and effective use of drugs and that may affect patient
43 outcomes is recommended" this seems to be a run-on sentence. Please rewrite.

line 47: why the community pharmacy setting is a barrier. please elaborate

line 47-51 - this is not a grammatically correct sentence. Are they examples?

line 52: dispensing, not dispensation

line 59: what do you mean not systematic?

line 62-65: very long sentence. please re-write

line 90: why this time period 2004-2015?

line 95: please elaborate which areas are the French speaking part of Switzerland

line 98: why were only 10 pharmacists selected? Your limitation paragraph acknowledge the small sample size, but it does not look like there are any efforts to improve it.

line 103: which are the 39 types of DRP? There are only 7 recognized categories of DRPs:
Strand LM, Morley PC, Cipolle RJ, Ramsey R, Lamsam GD (1990). "Drug-related problems: their structure and function". DICP. 24 (11): 1093–7. doi:10.1177/106002809002401114. PMID 2275235

line 143: some examples of the DRPs would be helpful. It helps readers to understand how serious some of the problems are

line 257-259: I don't think you can use this other small study to justify you can validate a tool with a small sample size. It is entirely possible that this other study validation method is equally problematic.

Author Response

October 28, 2019

Ruth Bai
Assistant Editor

Pharmacy MDPI

Subject:  Response to reviewers - Paper pharmacy_610695

We thank the editors and reviewers for their helpful comments and suggestions. In the following, the reviewers’ comments are reproduced integrally, numbered and appeared in bold text. Our response follows each comment. Any text cited from the manuscript has been put between quotation marks, while any addition to the original text is in italics; changes on response to the reviewer comments are in a redlined copy of the manuscript. Line numbers refer to those in the redline manuscript.  

Reviewer 2

Please find attached the letter fo response to the Editor an all the Reviewers

General comments:

There are several grammatical errors that make this manuscript very difficult to read. I pointed out only some of these errors below.

The manuscript was already submitted to American Journal Experts for revision before its submission. We made corrections for the comments listed below. In addition, a second revision has been asked in parallel to this revised version. It will be send to the editor as soon as possible.

Some of the examples given are very vague.

We have addressed the examples cited in the comments below.

It is difficult to validate the use of this tool with such a small sample size (n = 10).

 The sample size was based on the experience of previous studies on tools that had demonstrated at least a moderate level of agreement using a similar amount of raters. Highly variable reliability coefficients could be expected with a small sample size. Hence, for a better precision, we have added the confidence intervals for kappa coefficients in the Results section. These show that the level of uncertainty observed for the reliability coefficients was above the lower limit (deemed acceptable) for moderate agreement.

“Among the 240 documented cases, one was excluded for noncompliance with the documentation rules. Fleiss's kappa index values obtained for each step were above 0.40; for step 1a: 0.54 (95% confidence interval (CI): 0.53; 0.54); step 1b: 0.54 (95%CI: 0.51; 0.55); step 2: 0.57 (95%CI: 0.53; 0.61); step 3: 0.48 (95%CI: 0.43; 0.54); and step 4: 0.49 (95%CI: 0.47; 0.52).

Table 2. Results for test-retest reliability. Fleiss’s kappa index obtained for each participant”

Pharmacist Identification number (ID)

ID 1

ID 2

ID 3

ID 4

ID 5

ID 6

ID 7

ID 8

ID 9

ID 10

Kappa index (95%CI)

0.78

(0.65; 0.90)

0.58 (0.50; 0.66)

0.67 (0.60; 0.74)

0.68 (0.58; 0.74)

0.61 (0.54; 0.69)

0.53 (0.45; 0.61)

0.52 (0.44; 0.60)

0.59 (0.52; 0.66)

0.62 (0.54; 0.69)

0.57 (0.49; 0.65)

It is unclear whether this study received ethics approval.

The Swiss health research authority in the Canton of Vaud (Commission cantonale d'éthique de la recherche sur l'être humain- CER-VD), estimates that since this project is focused on the professional opinion of pharmacists regarding the use of the tool and only fictional cases, as featured in the vignettes (not real patient cases), were used the project is outside the application of the Swiss Law on Human research. Therefore, a formal submission including a protocol of the project and a Consent Form signed by participants were not required. This is now specified in the manuscript as follows:

[Lines 85-89] “The Swiss health research authority in the Canton of Vaud (Commission cantonale d'éthique de la recherche sur l'être humain- CER-VD), estimates that since this project is focused on the professional opinion of pharmacists regarding the use of the tool and as only fictional patients’ cases are used, the project is deemed to be outside the application of the Swiss Law on Human research.”

Specific comments:

line 38: "They" - do you mean DRPs?

Yes. As a result, we have reformulated the sentence to:

[Lines 38-42]: “DRPs can induce negative health and financial consequences.”

line 38-40: the examples here look very vague. I see that you have referenced. Can you go in details by what you mean by negative health and financial consequences? e.g. provide some figures.

We have added details to the introduction section. The text now reads as follows:

[Lines 39-42]: “DRPs can induce negative health and financial consequences. Stark et al. have estimated that DRPs may have accounted for 816 million Euros of health care expenditures over 1 year in Germany for 2.14 million ambulatory patients; with 80% of expenditures related to hospitalisations. In order to prevent such consequences, DRPs management needs to be optimized.”

Also, please elaborate what you mean by "context"

We were referring to the setting (community pharmacy). To avoid confusion, the sentences have been reworded as follows:

[Lines 43-45]: “The management of DRPs involves different clinical activities, mainly determined by the setting (hospital or community pharmacy). Likewise, the documentation of such activities is performed differently depending also on the setting.”

line 42-43: "to ensure the safe and effective use of drugs and that may affect patient
43 outcomes is recommended" this seems to be a run-on sentence. Please rewrite.

The text has been rewritten as follows:

[Lines 45-48]: “It has been recommended for pharmacists to document their professional activities that are intended to ensure the safe and effective use of drugs and which may affect patient outcomes”.

line 47: why the community pharmacy setting is a barrier. please elaborate

What we actually meant is that most of the tools are not adapted to pharmacists’ practice in the community pharmacy setting (mainly adapted to this specific workflow) and this constitutes an important barrier. To avoid confusion, the text “community pharmacy setting” was taken off the end of the sentence. The text now reads as follows:

[Lines 50-52]: “Community pharmacists face different barriers to documenting their clinical activities, particularly the lack of standardized documentation systems

adapted to the workflow inside pharmacies.”

line 47-51 - this is not a grammatically correct sentence. Are they examples?

Actually these were the reasons which make tools incompatible with the workflow in community pharmacy. The text has been rewritten as follows:

[Lines 53-57]: “The existing documentation tools have been deemed incompatible with the workflow in community pharmacy. Among the reasons for such incompatibility have been cited: the tools’ complexity; the omission of the actions taken by the pharmacist to resolve the DRP; a greater focus on the classification of the DRP rather than on the pharmaceutical intervention and its clinical significance; or inclusively, the time consumed to complete them”.

line 52: dispensing, not dispensation

The text has been rewritten:

“Furthermore, some clinical activities in community pharmacies often remain unacknowledged since the main role of community pharmacists has been drug dispensing”.

line 59: what do you mean not systematic?

We meant that the documentation is not performed regularly  and that it is not based on a structured approach. The text has been rewritten as follows:

[Lines 66-67]: “However, the documentation of such activities is not performed on a regular and structured basis in Swiss community pharmacies”.

line 62-65: very long sentence. please re-write

As requested, the text was rewritten;

[Lines 66-69]: “However, the documentation of such activities is not performed on a regular and structured basis in Swiss community pharmacies. Likewise, the information about DRP management process, DRP consequences or involved partners (other than the pharmacists and patients) is frequently missing”. 

line 90: why this time period 2004-2015?

The text has been rewritten to clarify this:

[Lines 110-111]: “2) published between 2004 and 2015 (to comprise the data published during ten years prior to this study”.

line 95: please elaborate which areas are the French speaking part of Switzerland

The text has been rewritten to be more specific:

[Lines 116-117]: “A prospective observational study was conducted in the regions of Geneva and Lausanne, Switzerland from April 1st to May 15th, 2015”.

line 98: why were only 10 pharmacists selected? Your limitation paragraph acknowledge the small sample size, but it does not look like there are any efforts to improve it.

The sample size was based on the experience of previous studies on tools that had demonstrated at least a moderate level of agreement using a similar amount of raters. As explained in our previous response on sample size, highly variable reliability coefficients could be expected with a small sample size. However, the observed results show an acceptable level of agreement. We have added further explanations to the Discussion section:

[Lines 302-310]: “Regarding the limitations of the study, the sample size can be considered relatively small (10 pharmacists). Reliability coefficients are dependent on the number of possible ratings, level of agreement among raters and difference among coefficients for hypothesis testing [52]. It is likely that the smaller the sample size is, the higher the variability for the coefficients would be observed. Therefore, a significant variability in kappa coefficients could be expected with this sample size. However, as shown by the observed values for kappa coefficients and their respective confidence intervals above 0.40, a level of moderate agreement among raters was determined. Hence ClinPhADoc can be considered as reliable to be used in clinical practice [25,53]. It is probable that a bigger sample would decrease the variability observed in reliability coefficients [53]”.

line 103: which are the 39 types of DRP? There are only 7 recognized categories of DRPs:
Strand LM, Morley PC, Cipolle RJ, Ramsey R, Lamsam GD (1990). "Drug-related problems: their structure and function". DICP. 24 (11): 1093–7. doi:10.1177/106002809002401114. PMID 2275235

The text was referring to the checkboxes in the tool. To avoid confusion, the text “divided in 39 types” has been removed and the sentence now reads as follows:

[Line 174]: “The steps included the following: 1a) 3 categories of DRP (clinical, technical and procedural); …”.

line 143: some examples of the DRPs would be helpful. It helps readers to understand how serious some of the problems are

The line 143 is the start line of the Results section in the Manuscript for revision. We wonder if the Reviewer refers to 1) the DRPs that are possible to document with the tool in current practice, 2) the DRPs that were used to test the reliability or 3) the DRPs documented individually by the raters. In any case, the type of DRPs correspond to all those applicable to the cases included in the tool. The seriousness/severity of documented DRPs (fictional or real) was not assessed. Indeed, the aim of this study was not to determine the prevalence and nature of DRP encountered in Swiss community pharmacies.

The evaluation of the seriousness or severity of the DRPs was not considered to be integrated as part of the tool. There are other already validated tools allowing the estimation of severity in different contexts such as NCC MERP (Taxonomy of Medication Errors. Available online: https://www.nccmerp.org/taxonomy-medication-errors-now-available), Dean et al. (Am J Health-Syst Pharm. 1999; 56:57–62) or the SCOPE criteria (Am J Health Syst Pharm 2015, 72, 1876-1884, doi:10.2146/ajhp140765). If needed, such tools could be used alongside to ClinPhADoc to eventually determine the prevalence according to the severity of DRP observed in Swiss community pharmacies.

line 257-259: I don't think you can use this other small study to justify you can validate a tool with a small sample size. It is entirely possible that this other study validation method is equally problematic.

As explained in our responses to the previous comments on sample size, we based our sample on the experiences of previous studies on tools showing reliability coefficients with a moderate level of agreement. As we have rewritten our text, the reference cited in this comment has been removed from the Discussion section. However, it would be problematic if the reliability coefficients were highly variable or under the limit of 0.40. For our study, we have added the 95% confidence intervals which show that even on the lower limit of the respective 95%CI, kappa coefficients are above the value of 0.40, which is deemed as moderate agreement.

Reviewer 3 Report

This paper reports on the development of a clinical documentation tool for DRPs management by Swiss community pharmacists. Overall it is very well written, and congratulations are due to the authors for such a nice work!

The comments bellow are intended to improve it further:

Abstract:

Clear and well structured. Minor corrections of written English are necessary (e.g. line 27 – “To optimize of the tool”  - remove “of”

Background:

Well written background section, providing the reader with all the information necessary about the addressed problem; clearly states the aims of the study

Methods:

Could use a figure to illustrate the 4 steps of ClinPhADoc. This would help the reader to quickly visualize the steps; Sampling strategy is adequate, but randomly selecting 10 pharmacists from this small group is only representative of the group. This seems more like a convenience sample (as the authors note in the “limitations part” of the discussion)

Results:

Line 176  - “Among tehse” – change to “Among these”

Results from the survey are interesting. 80% of participants agree that time is not an obstacle. At the same time, only 60% would like to use the tool in daily practice. As the authors note, this small group of participants are expected to be more willing to document their activities. If time is not an issue, what would be the reason for not wanting to use the tool? Is it only a remuneration issue as it seems to be intuited? Maybe this should also be addressed in future studies.

It seems part of the problems about documentation of pharmacists activities stated in the background (incompatibility with the work flow because of their complexity) are maintained in the document to be tested. Are the authors considering to “streamline” the online version of the tool for future dissemination?

Author Response

Reviewer  3

(Please find attached  the letter of response to the Editor and all the Reviewers)

Abstract:

Clear and well structured. Minor corrections of written English are necessary (e.g. line 27 – “To optimize of the tool”  - remove “of”

The manuscript was already submitted to American Journal Experts for revision before its submission. A second revision has been asked in parallel to this revised version. It will be send to the editor as soon as possible. Regarding line 27, the text in the manuscript now reads as follows:

[Line 27]: “To optimize the tool, pharmacists proposed developing an electronic version.”

Methods:

Could use a figure to illustrate the 4 steps of ClinPhADoc. This would help the reader to quickly visualize the steps

As requested, we have added a figure in the Results section. Please see the embedded figure in the attached letter.

“Figure 1. Visual representation of the steps in the ClinPhADoc tool”

Sampling strategy is adequate, but randomly selecting 10 pharmacists from this small group is only representative of the group. This seems more like a convenience sample (as the authors note in the “limitations part” of the discussion)

We agree that it is not possible to generalize the acceptability of the tool to a larger scale (i.e. acceptability by Swiss community pharmacists). Likewise, the variability in reliability coefficients might be higher with a small sample size. We have added further explanations to the limitations inflicted by a small sample size in the Discussion section:

Regarding the limitations of the study, the sample size can be considered relatively small (10 pharmacists). Since reliability coefficients are dependent on the number of possible ratings, the level of agreement among raters and the difference among coefficients for hypothesis testing [52]; it is likely that the smaller the sample size is, a higher variability would be anticipated for the coefficients. Therefore, a significant variability in kappa coefficients could be expected with this sample size. . However, as shown by the observed values for kappa coefficients and their respective confidence intervals above 0.40, a level of moderate agreement among raters was determined. Hence ClinPhADoc can be considered as reliable to be used in clinical practice [25,53]. It is probable that a bigger sample would decrease the variability observed in reliability coefficients [53]”.Furthermore, pharmacists were recruited on a voluntary basis rather than by random sampling; consequently, a selection bias cannot be completely excluded. Their incentive and willingness to document their clinical activities is expected to be better than for other pharmacists. In addition, participating pharmacists documented only clinical interventions related to DRPs they selected, rather than systematically. Therefore, the appropriateness and acceptability of the tool observed in this study might differ when used in a larger scale in real practice.”

Results:

Line 176  - “Among tehse” – change to “Among these”

The text now reads as follows:

“Among these activities, 5 were excluded for noncompliance with the documentation rules.”

Results from the survey are interesting. 80% of participants agree that time is not an obstacle. At the same time, only 60% would like to use the tool in daily practice. As the authors note, this small group of participants are expected to be more willing to document their activities. If time is not an issue, what would be the reason for not wanting to use the tool? Is it only a remuneration issue as it seems to be intuited? Maybe this should also be addressed in future studies.

For this study, participants were not asked the motives to use or not use the tool in their daily practice. Currently, Swiss pharmacists are not remunerated for documenting the DRPs they identify. In addition, they are anyway paid for their clinical activities (based on a fee-for-service) without having to document them. Therefore, such documentation is probably mostly perceived as a time consuming task offering no additional advantage for pharmacists. An additional statement has been added to the discussion:  

“The majority of pharmacists agreed on the importance of documenting their clinical activities and on the usefulness of the ClinPhADoc. However, only 60% of them attested that they would use it to document clinical activities in their daily practices. Participants were not asked the motives of such responses. In Switzerland, the documentation of identified DRPs in community pharmacy is not remunerated; nevertheless clinical activities are paid (based on a fee-for-service) without being documented. Lack of remuneration is known to be a common barrier in the usual practice of community pharmacists. Likewise, it has been reported that pharmacists perceive that their interventions increase their workloads without being to their advantage (other than providing a sense of professional accomplishment). A decrease in the documentation of DRPs and associated interventions as a result of lack of incentives was observed in a study conducted in 2008. This study showed that the mean rate of documented interventions decreased over the 4 weeks of the study, from 1.04% during the first week to 0.45% during the last week, in 15 of 20 pharmacies [17]. Therefore, even with an appropriate tool and with documentation not seeming to be time-consuming, documentation needs to be accomplished during a timeframe defined beforehand to maintain a steady reporting rate of pharmacists’ clinical activity.”

It seems part of the problems about documentation of pharmacists activities stated in the background (incompatibility with the work flow because of their complexity) are maintained in the document to be tested. Are the authors considering to “streamline” the online version of the tool for future dissemination?

As explained in the discussion section (lines 224-228), the mean duration of a clinical activity, including documentation, was estimated to be 6 minutes and 36 seconds, which was similar to the mean duration of clinical activity in pharmacies in Geneva and Lausanne (estimated to 6 minutes and 48 seconds by an external observer). Even though this was not a direct comparison, the documentation does not seem to add complexity to the pharmacy workflow. However, we agree that an electronic version of the tool may diminish the time and would be easier to use (as recommended by the study participants). As stated in the discussion section (lines 251-252) “An electronic version of the ClinPhADoc tool is currently under development.” Once this electronic version is optimized, the CinPhADoc tool will be proposed to be used in other community pharmacies.

Round 2

Reviewer 2 Report

I see that the authors have addressed my earlier comments.

My main concern is the significance and scientific soundness of the study because it is a small study (n = 10) with limited implications.

I do not have anything else to add.

Author Response

November 4th, 2019

Ruth Bai
Assistant Editor

Pharmacy MDPI

Subject: Response to reviewers - Paper pharmacy_610695

Dear Mrs Bai,

Further to your decision on manuscript pharmacy_610695, we are pleased to submit a new revised version of our paper.

We thank the editors and reviewers for their helpful comments and suggestions. The most recent comments of Reviewers are reproduced integrally, and appear in bold text. Our response follows each comment.

[Please find attached the letter of response to the Editor and Reviewers]

Reviewer 2

I see that the authors have addressed my earlier comments.

My main concern is the significance and scientific soundness of the study because it is a small study (n = 10) with limited implications.

I do not have anything else to add.

The small sample size does not constitute a threat to the scientific validity of our results. Reliability tests have been usually carried out with 2 raters with more clinical cases (at least 30 cases) than those we have used and succeeded in obtaining highly reliable tools. For instance, Snyder et al (Pharmacoepidemiol Drug Saf. 2007 Sep;16(9):1006-13) obtained a kappa from 0.66 (95% confidence interval [CI95%]: 0.58–0.76) to 0.84 (CI95%: 0.74–0.93) using 2 raters for testing the reliability of the NCC MERP classification for medication errors. Also with 2 raters, Forrey et al. (Am J Health Syst Pharm. 2007 Jan 15;64(2):175-81.) obtained a kappa of 0.61 (CI95%: 0.41-0.81). More recently, Desrochers et al. (Am J Kidney Dis. 2011 Oct;58 (4):527-35.) tested the reliability of the PAIR criteria with 2 raters obtaining a kappa of 0.80 (95% CI, 0.72-0.87). In our study, we increased the number of raters and used 21 cases. We provided the confidence intervals of kappa coefficients as a more precise measure of our estimate. Confidence intervals indicate the level of uncertainty, which is particularly relevant when a study recruits only a small sample of the overall population. By having an upper and lower confidence limit, we can infer that the true population effect is included between these two points.

As we explained in the article, with this sample size we succeeded in obtaining a kappa that is already above the limit considered as reliable. It is likely that a bigger sample size would shorten the confidence intervals, meaning that the estimate would be more precise. But even on the lower limit, the estimated kappa is deemed reliable with this sample size.
